# Antibody-Drug Conjugates for Breast Cancer Treatment: Emerging Agents, Targets and Future Directions

**DOI:** 10.3390/ijms241511903

**Published:** 2023-07-25

**Authors:** Tinglin Yang, Wenhui Li, Tao Huang, Jun Zhou

**Affiliations:** Department of Breast and Thyroid Surgery, Union Hospital, Tongji Medical College, Huazhong University of Science and Technology, Wuhan 430022, China

**Keywords:** antibody-drug conjugates, targeted therapy, breast cancer

## Abstract

To achieve the scheme of “magic bullets” in antitumor therapy, antibody-drug conjugates (ADCs) were developed. ADCs consist of antibodies targeting tumor-specific antigens, chemical linkers, and cytotoxic payloads that powerfully kill cancer cells. With the approval of ado-trastuzumab emtansine (T-DM1) and fam-trastuzumab deruxtecan (T-DXd), the therapeutic potentials of ADCs in breast cancer have come into the spotlight. Nearly 30 ADCs for breast cancer are under exploration to move targeted therapy forward. In this review, we summarize the presenting and emerging agents and targets of ADCs. The ADC structure and development history are also concluded. Moreover, the challenges faced and prospected future directions in this field are reviewed, which give insights into novel treatments with ADCs for breast cancer.

## 1. Introduction

In cancer treatment, maximizing the injury of tumor cells while reducing the damage to normal tissues has always been the pursuit. As the normal adjuvant therapy, chemotherapy may cause various severe side effects during treatment due to its relatively low specificity [1]. The scheme of “magic bullets” was first imaged in the early 20th century, describing the direct access of some compounds to specific targets within cells that could cure the disease [2]. For antitumor therapy, the ideal agents should give precision strikes to only cancer cells and be innoxious to normal tissues. Despite great progress with a series of monoclonal antibodies targeting antigens that are aberrantly expressed in tumors in recent decades, they have shown insufficiency in certain cases [3].

To achieve the goal of ideal treatment, antibody-drug conjugates (ADCs) that combine antibodies and cytotoxic drugs with a linker have been developed. ADCs combine the advantages of chemotherapy and monoclonal antibodies, composing the ability to reach tumors precisely and with elevated tumor-killing toxicity [4]. For drug resistance to targeted therapy with monoclonal antibodies, the activation of alternative pathways is a common mechanism [5]. Compared with the strategies of combining antibodies, ADCs that facilitate both antibody-mediated effects and cytotoxic payloads provide preferable efficacy by the addition of different antitumor effects. Clinical trials of ADCs in tumor treatment confirmed markable therapeutic effects, leading to the launch of Mylotarg (gemtuzumab ozogamicin), the first ADC to be approved by the Food and Drug Administration (FDA) since 2000 [6]. There have been 15 ADCs that received market approval so far worldwide, and over 100 ADC candidates are being investigated in clinical stages at present [7].

As the most common cancer worldwide, breast cancer is of great clinical interest, and its treatment is being revolutionized by the emergence of ADCs [8]. Based on the success of anti-human epidermal growth factor receptor 2 (HER2) antibodies in HER2-positive breast cancer, anti-HER2 ADCs were explored and then constructed. Ado-trastuzumab emtansine (T-DM1) and fam-trastuzumab deruxtecan (T-DXd) were subsequently approved by the FDA for the treatment of HER2-positive metastatic breast cancers [9,10]. In further exploration, an anti-trophoblast cell surface antigen 2 (Trop2) ADC named sacituzumab govitecan (IMMU-132, Trodelvy, SG) was also approved for treating triple-negative breast cancers (TNBCs) [11]. With expanding novel targets and indications, ADCs are expected to lead a new era of targeted cancer therapy in breast cancer. In this article, ADCs for breast cancer treatment are reviewed, including existing and emerging agents, targets, challenges faced in the field, and future directions for ADC development. Representative clinical trials are discussed, as well as the structure and evolution of ADCs.

## 2. Structure and Evolution of ADCs

### 2.1. The Structure of ADCs

ADCs consist of three elements: antibodies, chemical linkers, and cytotoxic payloads (Figure 1) [7].

Antibodies are responsible for recognizing tumors, for which the choice of target antigens plays a pivotal role. The target antigen is the direction guidance for ADCs to neoplasms, which are also attributed to the internalization of cytotoxic payloads into cancer cells. An ideal antigen should be expressed only or primarily in tumor cells, be non-secretory, and can be internalized when binding with the matching antibodies [12]. Conventionally, molecules that are specifically highly expressed in cancer cells are often selected as antigens, and the possibility of targeting potential antigens in the tumor microenvironment (TME) is coming to the fore [13]. To precisely direct ADCs to cancer cells, antibodies need to be facilitated with high affinity, appropriate plasma half-life, as well as efficiency to enter cells [14]. To maintain low immunogenicity, humanized antibodies were put into ADC construction [15].

Linkers are the significant component deciding the stability of ADCs and the effective release of payloads in cancer cells. The mainstream of linkers includes cleavable and non-cleavable linkers, depending on the metabolic outcomes in cells [16].

Cytotoxic payloads are the part that actually kills tumor cells in ADCs. Also known as a warhead, payloads are usually small molecules with high efficacy, including tubulin inhibitors, DNA-targeting agents, and immunomodulators [17].

Together, ADCs play an antitumor role through the following steps (Figure 2). Step 1: circulating and binding antigens. ADCs are mostly administered intravenously to enter circulation, after which antibodies recognize and combine with antigenic epitopes specifically. Step 2: internalization. Endocytosis would begin in the tumor cell after antigen-antibody binding, making the ADCs swallowed into the cell. Step 3: drug release and tumor eradication. Of note, the active cytotoxic load can also kill surrounding cancer cells, known as the bystander effect [18].

### 2.2. First-Generation ADCs

First-generation ADCs, represented by Mylotarg, were not satisfying due to the high off-target toxicity and low drug efficacy. Mylotarg was designed to treat leukemia, in which calicheamicin was taken as a cytotoxic payload to cause DNA damage. Calicheamicin was linked to gemtuzumab, a mutated anti-CD33 IgG4 subtype monoclonal antibody with a cleavable linker containing a hydrazone bond [19]. Although hydrazone should be stable at physiological pH, in theory, Mylotarg exhibited instability in circulation that led to its off-target toxicity. Moreover, the conjugation of antibodies and payloads of first-generation ADCs was based on the random coupling of amino acid residues, resulting in a highly heterogenous drug-to-antibody ratio (DAR) mixture and fluctuating efficacy [20].

### 2.3. Second-Generation ADCs

Second-generation ADCs, represented by Brentuximabvedotin and T-DM1, have improved clinical efficacy and safety. DM1 was coupled to the lysine residue of trastuzumab via a non-cleavable linker, succinimidyl 4-(*N*-maleimidomethyl)cyclohexane-1-carboxylate (SMCC) [21]. Since the linker was non-cleavable, T-DM1 could turn to an active state only by digestion in the lysosome after internalization. T-DM1 was an effective drug for HER2-positive breast cancer; hence, it was approved by the FDA in 2013 [22]. Linker improvements in second-generation ADCs achieved better plasma stability and harmonized DAR distribution. The cytotoxicity and coupling of payloads in second-generation ADCs were also improved, and the increase of payloads’ water solubility reduced the occurrence of antibody aggregation.

In addition, some of the second-generation ADCs utilized cleavable linkers, such as Adcetris and Polivy for lymphoma treatment [23,24]. These ADCs with cleavable linkers can cause a bystander effect in vivo, referring to the phenomenon that payloads can be released and kill surrounding cells. Despite the considerable safety advances of second-generation ADCs, the shadow therapeutic window due to off-target toxicity is still a challenge for ADCs [25].

### 2.4. Third-Generation ADCs

Well-known third-generation ADCs include T-DXd, SG, and later-approved ADCs [26,27]. Compared to first and second-generation ADCs, third-generation ADCs have a more harmonious DAR, showing less off-target toxicity. Antibodies used in third-generation ADCs are humanized antibodies rather than chimeric antibodies, leading to a reduction in immunogenicity [28]. Fab fragments of antibodies are in exploration, which helps the internalization of ADCs. Additionally, multiple types of payloads like immunomodulators are in application [4]. Third-generation ADCs exhibited better antitumor efficacy, lower toxicity, and a more proper half-lifetime in circulation.

Next-generation ADCs are putting efforts into combining different targets or payloads to enhance specificity and curable effects. Meanwhile, developing non-internalized ADCs and further reducing molecular weights of ADCs are also research hotspots [29].

## 3. ADCs Targeting HER2 in Breast Cancer

The positive expression of HER2 has been established as a biomarker for molecular typing of breast cancer. Abnormal expressions of HER2 regulate a series of pathways and promote tumor cell proliferation. In the field of treating HER2-positive breast cancers, trastuzumab has achieved monumental success. Based on trastuzumab and other antibodies against HER2, various ADCs were constructed targeting HER2 [30]. Six representative anti-HER2 ADCs are discussed as follows, as shown in Table 1.

### 3.1. T-DM1

T-DM1 is the first ADC targeting HER2, which is developed by linking DM1 to trastuzumab with a non-cleavable linker. The phase III EMILIA study (NCT00829166) evaluated T-DM1 in 991 HER2-positive advanced breast cancer patients who have experienced trastuzumab and taxane treatment. Compared with the capecitabine plus lapatinib group, the T-DM1 group improved progression-free survival (PFS) by 3.2 months and overall survival (OS) by 5.8 months [9]. Based on positive responses in advanced breast cancers, the KATHERINE study (NCT01772472), a phase III trial was set to test T-DM1 as adjuvant therapy for patients with residual tumors. Invasive disease-free survival (iDFS) in the T-DM1 group was 88.3%, significantly more than the 77% iDFS rate in the trastuzumab group [31]. The ATEMPT trial (NCT01853748) compared T-DM1 with paclitaxel plus trastuzumab in Stage I HER2-positive breast cancer patients. The 3-year iDFS for the T-DM1 group reached 97.8%, exhibiting an exciting outcome without an obvious increase in adverse events [32]. Based on promising results from plenty of clinical trials, the T-DM1 has been approved for second-line HER2-positive metastatic breast cancer treatment and therapy for HER2-positive early-stage breast cancer with residual cancer following neoadjuvant taxane and trastuzumab [33]. Clinical trials are on the way to explore the efficacy of T-DM1 as neoadjuvant therapy or first-line therapy.

An important field in ADC drug application is combination regimens, and immunotherapy is in rising exploration. Immunotherapy is a burgeoning therapy that eradicates tumor cells by stimulating the immune system of the host, among which the immune checkpoint inhibitors (ICIs) are the most widely-used immunotherapy in solid tumors. Emerging evidence has shown that ADCs contribute to enhancing antitumor immunity and may conduce to overcoming the resistance of immunotherapy [34]. T-DM1 exhibited the ability to enhance T-cell responses in the combination of ICIs in preclinical models, emphasizing the potential to be a candidate in clinical practice [35]. The phase II KATE2 study (NCT02924883) is the first clinical trial that explored the combination of T-DM1 and atezolizumab. Atezolizumab is an ICI against programmed cell death receptor 1 (PD-1), one of the most successful immune checkpoints. In the KATE2 study, 202 enrolled participants with HER2-positive locally advanced or metastatic breast cancer who have previously received trastuzumab and taxane were treated with T-DM1 plus atezolizumab or T-DM1 plus placebo at a ratio of 2:1. The median PFS was 8.2 months for atezolizumab group, and 6.8 months for the placebo group (*p*-value = 0.33). The difference in median PFS among programmed cell death ligand 1(PD-L1) -positive population was not significant, either (8.5 months vs. 4.1 months, *p*-value = 0.10) [36]. Although T-DM1 plus atezolizumab did not provide meaningful clinical improvement in the KATE2 study, further studies on the combination of ADCs and ICIs are on the way. The ongoing phase III KATE3 study aimed to recruit 96 participants with locally advanced or metastatic HER2-positive breast cancer. The PFS and OS will be measured in the T-DM1 plus atezolizumab group and the T-DM1 plus placebo group. Another phase III study, Astefania (NCT04873362), was set to explore combining T-DM1 with atezolizumab for HER2-positive breast cancer patients at high risk of recurrence. T-DM1 with atezolizumab or placebo would be given to patients following preoperative therapy, and iDFS was set as the primary endpoint. In this ongoing study, trastuzumab would be used to complete 14 treatment cycles if T-DM1 treatment was terminated due to toxicity [37].

### 3.2. T-DXd

T-DXd was constructed with trastuzumab, cleavable linkers, and topoisomerase inhibitors named deruxtecan (DXd). The DESTINY breast-01 study (NCT03248492) confirmed the efficacy of T-DXd. The study consists of two parts in patients with HER2-positive metastatic breast cancer who has received T-DM1 treatment. A recommended dose of 5.4 mg/kg was established in part one, and the dose was administered to participants in part two. The objective response rate (ORR) was 60.9% in the intention-to-treat population, and 6% of patients achieved a complete response. The median PFS was 16.4 months, with a mean response duration of 1.6 months. Compared with chemotherapy, T-DXd achieved antitumor activity after a similar rapid response duration [38]. The finding accelerated the exploration of T-DXd, both in metastatic second or third-line treatment and in first-line treatment. The effects of T-DXd and T-DM1 for the treatment of HER2-positive advanced patients who have received trastuzumab and taxane were compared in the clinical trial DESTINY breast-03 (NCT03529110). A total of 584 patients were randomized in this study. The 12-month PFS was 75.8% for the T-Dxd group and 34.1% for the T-DM1 group. The OS was 94.1% and 85.9%, respectively [39]. The PFS and OS interpreted a favorable efficacy of T-DXd. In addition, the potential of T-DXd as an adjuvant therapy was accessed in the trial DESTINY breast-09 (NCT04784715). T-DXd alone, T-DXd plus pertuzumab, and the combination of trastuzumab, pertuzumab, and taxane are three arms of DESTINY breast-09. The PFS will be observed and compared.

Of note, T-DXd exhibited certain efficacy in HER2-low breast cancer in a phase I study (NCT02564900) [40]. This remarkable property of T-DXd was further investigated in DEATINY breast-04 study (NCT03734029). Patients with HER2-low breast cancers who have experienced 1 or 2 prior chemotherapies were enrolled in this phase III study. The median PFS for the T-DXd group was 10.1 months, while it was 5.4 months for the physician’s choice group. Despite the predominant efficacy T-DXd has achieved, it may be allied with higher grades of adverse events. The toxicity of T-DXd should be monitored and managed while utilizing its antitumor activity [41].

Similarly, the efficacy of combining T-DXd with immunotherapy was investigated. Durvalumab is a monoclonal antibody belonging to ICIs that is against PD-L1. The combination of T-DXd and durvalumab was explored in module 1 of the DESTINY breast-07 study (NCT04538742) and module 2 of the DESTINY breast-08 study (NCT04556773). These two studies were designed to assess the safety, tolerability, and antitumor activity of T-DXd plus durvalumab in HER2-positive and HER2-low advanced or metastatic breast cancers, respectively. Occurrences of adverse events and serious adverse events would be monitored. In addition, PFS and ORR would be measured as secondary endpoints to preliminarily evaluate the efficacy. Other combination regimens with T-DXd, such as chemotherapy drugs, targeted therapy drugs, and endocrine therapy drugs would also be assessed in these two studies.

### 3.3. Other ADCs Targeting HER2

A range of other ADCs targeting HER2 are in trials, including RC48, SYD985 (trastuzumab duocarmazine), A166, and ARX788.

RC48 is made up of microtubule inhibitor payload monomethyl auristatin E (MMAE) and a humanized monoclonal antibody against HER2, hertuzumab. RC48 has been approved for advanced HER2-positive gastric cancer and urothelial carcinoma by the FDA and National Medical Products Administration (NMPA), boosting its exploration in breast cancers. A phase I study (NCT02881190) tested the efficacy of RC48 at 2.0 and 2.5 mg/kg for patients with solid tumors, which both showed 33.3% ORR and 53% disease control rate (DCR). The safety profile was also manageable [42]. RC48 showed an ideal DCR of 96.7% (29 among 30 patients with HER2-positive metastatic breast cancer) in a phase Ib study (NCT03052634). The dose of 1.5 mg/kg and 2.0 mg/kg were accessed in this study, leading to 26.7% and 46.7% ORR, respectively [43].

SYD985 combines duocarmycin and the backbone of trastuzumab with a cleavable linker. The payload duocarmycin is cell permeable, which enables SYD985 to kill surrounding cells at whatever level HER2 was expressed. SYD985 showed clinical activity in previously heavily treated HER2-positive breast cancer, including those who were resistant to T-DM1 [44]. The phase III TULIP trial (NCT03262935) aimed to further explore SYD985 or physician’s choice as second-line therapy. The PFS was improved by SYD985 for 2.1 months (7.0 months vs. 4.9 months), indicating the therapeutic potential of SYD985 in breast cancer patients who have received heavy treatments [45].

A166 comprises trastuzumab and microtubule inhibitor duosatatin-5. In the trial NCT03602079, the first-in-human phase I study of A166, 27 patients with solid tumors received A166 monotherapy. Of the 27 participants, 7 patients reached partial response. A166 showed meaningful efficacy in breast patients, for which DCR was 100% [46].

ARX788 is an ADC utilizing an anti-HER2 monoclonal antibody with the modified heavy chain Ala114 [47]. Results from a phase I clinical study (NCT03255070) revealed ARX788 can lead to the eradication of tumors, in which ORR ranged from 56% to 63% with increased doses. Ongoing phase II trials were set to explore the efficacy of ARX788 in both HER2-positive and HER2-negative metastatic breast cancer patients. A single-arm phase II study (NCT04829604) aimed to investigate ARX788 in HER2-positive metastatic breast cancer patients who have received T-DXd previously. The ORR will be monitored in 71 participants for 2 years. Another phase II study (NCT05018676) is recruiting patients with low expression of HER2. This study was set to observe the ORR of ARX788 in 54 participants.

## 4. Emerging Targets of ADCs in Breast Cancer

With the great progress of ADCs targeting HER2, efforts were taken to extend the application of ADCs in other subtypes of breast cancers. However, TNBC has long been known as a hard-to-cure subtype due to the lack of molecular markers [48]. For hormone receptor (HR)-positive breast cancers, endocrine therapy, rather than targeted therapy was recommended as the first-line therapy [49]. It was in doubt whether there existed potential targets for ADC treatment in HR-positive breast cancers and TNBCs. To address the efficacy of ADCs in these subtypes, as well as explore the possibility of improving therapeutic effects in HER2-positive breast cancers, more targets are being investigated in breast cancer. Herein, we discussed emerging ADC targets including Trop2, protein tyrosine kinase 7 (PTK7), receptor tyrosine kinase-like orphan receptor 1 (ROR1), ROR2, B7-H3, glycoprotein-NMB (gpNMB), Folate receptor α (FRα), HER3, and CA6. Representative ADCs and clinical trials are listed in Table 2.

### 4.1. Trop2

Trop2 is a transmembrane protein encoded by the TACSTD2 gene. With aberrant high expressions in tumor cells and low or no expressions in normal cells, Trop2 also regulates several signaling pathways [50]. Trop2 involves in promoting cell proliferation, inhibiting cell apoptosis, and regulating transcription factors via the MAPK pathway, β-catenin pathway, etc. Trop2 is overexpressed in about 62.4% of breast cancers and was especially overexpressed in TNBC at a rate of 78.1% [51]. Administering ADCs targeting Trop2 is another prospective treatment for breast cancer.

SG is an ADC targeting Trop2 with SN-38 payloads. In the phase III ASCENT study (NCT02574455), a total of 468 patients with locally advanced or metastatic TNBC were recruited. All participants received two or more chemotherapy regimens and did not have brain metastases before administering SG. There were 235 patients randomized to the SG group and 233 patients randomized to the chemotherapy group. For the refractory or relapsed participants, the median PFS in the SG group was 3.9 months longer than the chemotherapy group (5.6 months vs. 1.7 months, *p*-value < 0.001). The median OS was also improved by SG with 5.4 months (12.1 months vs. 6.7 months). The ORR was 35% for the SG group, while only 5% for the chemotherapy group [52]. Owing to the efficacy SG has shown, it has currently been approved by the FDA for the treatment of metastatic TNBC [53].

Another trial TROPiCS-02 (NCT03901339) aimed to study the effect of SG in HR-positive/HER2-negative breast cancer patients. Of 543 endocrine-resistant metastatic breast cancer patients in this phase III study, 272 patients received SG, and the rest 271 patients received chemotherapy. A longer median PFS was seen in the SG group (5.5 months vs. 4.0 months), and improved PFS rates at 6 and 12 months were also revealed (46% vs. 30%, 21% vs. 7%, respectively) [54].

Datopotamab Deruxtecan (Dato-DXd, DS-1062a) is another ADC targeting Trop2, which loads cytotoxic drug DXd. The phase I study TROPION-PanTumor01 (NCT03401385) was carried out in metastatic TNBC with failed previous treatment. The latest outcome of TROPION-PanTumor01 came up with a 32% ORR in the Dato-DXd group, of which one participant reached complete remission (CR). Encouraging efficacy and acceptable side effects motivated researchers to further access Dato-DXd in phase III clinical trials, including TROPION Breast01 (NCT05104866) and TROPION Breast02 (NCT05374512) [55].

SKB-264 is also an ADC targeting Trop2, utilizing novel topoisomerase I inhibitors as payloads. Preclinical experiments denoted the promising antitumor ability of SKB-264 and longer half-life compared with SG [56]. Trials administering SKB-264 are also recruiting locally advanced metastatic or recurrent TNBC patients in China (NCT05347134, NCT05445908, NCT04152499).

### 4.2. PTK7

The transmembrane PTK7 is a receptor tyrosine kinase in the Wnt signaling pathway, which was found to be highly expressed in TNBC and non-small cell lung cancer (NSCLC) [57]. Cofetuzumab pelidotin is an anti-PTK7 ADC coupled with a microtubule inhibitor (Au0101). A phase I clinical trial was set to explore the efficacy and safety of cofetuzumab pelidotin in advanced solid tumors (NCT02222922). In TNBC patients, curative effects were observed in 21% of the participants. Patients whose tumors expressed higher PTK7 may get better results with cofetuzumab pelidotin [58].

### 4.3. ROR1 and ROR2

ROR1 and ROR2 were identified as transmembrane proteins, acting as receptors of Wnt5a. High expression of ROR1 and ROR2 was proved to be associated with tumor invasion and proliferation, more rapid disease progression, as well as poor prognosis [59].

ROR1 was found to be highly expressed in tumors and limited expressed in normal tissues. Furthermore, ROR1 was detected to be relevant to important genes and pathways of TNBC, including the SMAD pathway and Wnt pathway. An ADC targeting ROR1 named NBE-002 was constructed. On the basis of the preclinical antitumor activity of patient-derived xenografts (PDX) models [60], the NBE-002 was being explored in an ongoing phase I/II clinical trial (NCT04441099). NBE-002 would be given to patients with advanced solid tumors, including TNBC intravenously. As primary endpoints, the recommended dose and antitumor activity of NBE-002 would be analyzed.

The expression of ROR2 was found to be relevant to the expression of ROR1, demonstrating its potential as an ADC target [61]. The tolerability and efficacy of CAB-ROR2-ADC targeting ROR2 were investigated in a phase I ongoing study (NCT03504488). A total of 420 patients with metastatic, locally advanced, or unresectable solid tumors including TNBC were recruited and received CAB-ROR2-ADC alone or combined with anti-PD1 reagents.

### 4.4. B7-H3

B7-H3, also known as CD276, is a molecule of the B7 superfamily, whose overexpression was correlated with cancer progression. Being highly expressed in various solid tumors and TME, B7-H3 also acts as an immunomodulator that inhibits T cell responses [62]. Enoblituzumab, an antibody against B7-H3, was combined with duocarmycin payloads to construct the ADC MGC018. MGC018 demonstrated curative effects in PDX models including TNBC, prostate cancer, and head and neck squamous cell carcinoma (HNSCC) [63]. The tolerance and pharmacodynamics of MGC018 were also favorable in cynomolgus monkeys. Therefore, an ongoing clinical trial was carried out to evaluate MGC018 in solid tumors (NCT03729596). This study aimed to explore the antitumor activity and dose of MGC018 in TNBC, NSCLC, HNSCC, and melanoma, and adverse events and toxicities would be primary endpoints.

### 4.5. gpNMB

The gpNMB is a glycoprotein expressed in multiple tumors and is linked with cancer promotion. Experimental evidence has shown the immunosuppressive role of gpNMB by ameliorating the activities of adaptive immune cells [64]. The gpNMB-targeted ADC is CDX-011 (glembatumumab vedotin, GV), which contains tubulin inhibitor payloads MMAE. The phase II EMERGE study (NCT01156753) and a phase I/II study (NCT00704158) investigated the CDX-011 in the treatment of breast tumors and found the safety profile was acceptable [65]. Additionally, the ORR in the CDX-011 group was 18% and reached 40% in TNBC patients [66]. Subsequently, the METRIC study (NCT01997333) was conducted to compare capecitabine with CDX-011 treatment in gpNMB-overexpressing TNBC patients. The median PFS was 2.9 months in the CDX-011 group and 2.8 months in the capecitabine group (*p*-value = 0.761), which showed no priority for CDX-011 [67]. Whether CDX-011 can be an effective treatment for TNBC needs further exploration.

### 4.6. FRα

Antitumor treatments interrupting the metabolism of folate have gained thriving progress. FRα is a receptor located on the cell surface that facilitates folate intake. The overexpression of FRα was detected in a variety of solid tumors and was associated with adverse outcomes [68]. The ADC MORAb-202 was developed, including a humanized farletuzumab against FRα, and eribulin was coupled as payloads. The administration of MORAb-202 exhibited elevated antitumor activity in preclinical TNBC PDX models and FRα-positive cell-derived xenograft (CDX) models [69,70]. An ongoing phase I/II study aimed to evaluate the safety and tolerability of MORAb-202 in endometrial cancer, ovarian cancer, TNBC, and NSCLC (NCT04300556). The study warranted enrolling 55 participants, and MORAb-202 will be intravenously administered. As primary outcomes, tolerance dose, ORR, toxicities, and adverse events will be monitored.

### 4.7. HER3

HER3 is a member of the HER family, whose functions rely on forming homodimerization with the HER family members. HER3 mainly homodimerizes with HER2 or EGFR, enhancing tumor proliferation, aggressive phenotypes, and drug resistance [71]. Various malignancies overexpress HER3, such as tumors in the digestive, urinary, and reproductive systems [72]. Patritumab deruxtecan (HER3-DXd, U3-1402) is an ADC targeting HER3, which contains anti-HER3 antibody patritumab, cleavable linker, and cytotoxic payload deruxtecan. A total of 184 patients with HER3-positive metastatic breast cancer were enrolled in a phase I/II study to investigate the dose and pharmacodynamics of HER3-DXd (NCT02980341). For the HER3-high/TNBC group, the ORR was 22.6%, and the ORR for HER3-high/HER2-positive groups reached 42.9%. Adverse events are neutropenia, thrombocytopenia, and anemia [73]. The efficacy of HER3-Dxd was also evaluated in early breast cancers. A window-of-opportunity study (NCT04610528) was conducted in HR-positive/HER2-negative breast cancer patients or TNBC patients. The ORR for the overall population was 32%, with TNBC and HR-positive/HER2-negative having ORRs of 35% and 30%, respectively. HER3-DXd for the neoadjuvant treatment of HER2-negative early breast cancer showed a promising ORR, which was independent of baseline HER3 levels, and the treatment is well-tolerated [74].

### 4.8. CA6

Cancer-induced aberrant glycosylation produces the CA6 sialoglycotope of Mucin-1 (MUC1)-glycoprotein. CA6 is barely observed in normal adult tissues, while it is often positively expressed in malignancies. The anti-CA6 ADC consists of DS6 monoclonal antibody, cleavable linker, and tubulin inhibitor DM4 [75]. The efficacy and safety of SAR566658 were investigated in CA6-positive metastatic TNBC patients (NCT02984683). Unfortunately, the study was terminated due to limited efficacy. In each arm, about 70% of patients encountered disease progression. The rate of adverse events ranged from 25% to 27%. Despite the antitumor activity SAR566658 has shown in preclinical PDX models, unsatisfying results in clinical research reminded it may not be an ideal treatment for TNBC.

The approval of SG for TNBCs provoked attention to expanding ADC treatment for breast cancers beyond the HER2-positive subtype. Of note, ADC treatment for these patients was permitted as second or third-line therapy. Additionally, the side effects of ADCs should be closely monitored when administered to advanced breast cancer patients.

## 5. Challenges and Future Directions in ADCs

With the in-depth understanding of the mechanism of ADCs and the continuous breakthrough of drug development technology, the new generation of ADCs has made progress. However, challenges are still faced and efforts are still warranted to promote the application of ADCs in frontline cancer treatment.

### 5.1. Challenges in ADCs

In clinical practice, the most obvious and unavoidable challenge was adverse effects. Target selection, drug mechanisms, chemistry characters of linkers, and coupling sites are all important determinants of ADC-related adverse effects. Severe adverse effects can occur in the hematologic system, liver, kidney, and lung, and adverse effects need to be dealt with in time [76]. Neutropenia, thrombocytopenia, leukopenia, and anemia may be related to the early release of cytotoxic payloads. Those cytotoxic drugs mainly injure the actively proliferating cells, which locates in the bone marrow and liver. In clinical settings, doses of ADCs need to be adjusted cautiously when encountering adverse effects. For hematologic toxicity, blood cell count ought to be well-monitored. Also, symptomatic drugs can be used to raise neutrophil, platelet, or erythrocyte counts. The immune response can be stimulated by ADC drugs, which may cause autoimmune responses, resulting in kidney injury [77]. Fatal lung injury manifests as interstitial lung disease (ILD) and was observed in patients receiving T-DXd [78]. The mechanism underlying ILD remains unclear, and one of the hypotheses was that abundant blood flow brought ADCs and payloads to the lung [41]. Once the ILD was diagnosed, the dose of T-DXd should be strictly controlled, and glucocorticoids should be administered. For severe circumstances, ADC treatment needs to be terminated. As the most widely used ADCs in breast cancer, specific adverse effects of T-DM1, T-DXd, and SG are listed in Table 3, as well as their frequencies. Although all anticancer drugs have adverse effects, seeking to minimize the toxicity to healthy tissues is always in pursuit.

Another major challenge specific to ADCs was drug efficiency which could be influenced by multiple factors. To precisely target tumor cells by intravenous injection, the half-life, and linkers were crucial. Proper half-life guarantees ADC to reach tumors [81], and the strength of the linker needs to be modified to avoid the early release of cytotoxic payloads. Internalization is also important to display drug efficacy [29]. Hence, expression levels of antigens and molecular weight of ADCs need to be arranged. Furthermore, since only a small fraction of ADCs can currently reach the tumor site, the toxicity of payloads should be powerful while avoiding toxicity.

Drug resistance, however, was encountered while the promising effect ADCs has been achieved. Drug resistance can be both primary (resistant at the beginning) and secondary (resistant after responding). Reasons for drug resistance can be antibody-related, payload-related, and cancer cell function-related. Inadequate antigen expression levels can lead to the deduction of ADC specificity, and mutated targets of payloads may also induce failure in therapy [82]. For cancer cells, impaired trafficking and destroyed lysosome function contribute to drug resistance [83,84]. Failure and disruption in each part of exploiting the antitumor activity can bring drug resistance.

### 5.2. Future Directions

The three key elements of ADCs are antibodies, linkers, and payloads. The direction of improvement also lies in modifying the three parts.

The concept of dual-epitope or dual-target ADCs has inspired ADC innovations. In breast cancer, there are candidates to be targeted together with HER2. By bispecific antibodies targeting HER2 and prolactin receptors (PRLR), the internalization and traffic efficiency were significantly improved, which promotes efficacy [85]. By combining HER2 and CD63 targets, the internalization of ADCs was enhanced in preclinical models, since CD63 is a membrane protein expressed in the lysosome [86].

Similarly, dual-payload was proposed in advancing ADCs. Greater efficacy can be realized by mixing two payloads at a fixed ratio and killing cancer cells synergistically. Payloads that kill cancer cells by different pathways also ameliorated the rate of drug resistance [87]. Therapies currently in development have been investigating these possibilities. The dual-payload ADC coupled with MMAE and monomethyl auristatin F (MMAF) showed antitumor activities in mouse models and brought light to overcome tumor heterogeneity [88].

Another strategy is to reduce the molecular weight of ADCs. The currently commonly used antibodies for constructing ADCs are IgG molecules, with an approximate molecular weight of 150 kDa. Replacing the antibodies with polypeptides to construct ADCs was proposed, which was expected to shrink the molecular weight and improve drug penetration. According to this strategy, the anti-somatostatin receptor 2 ADC PEN-221 was developed. The molecular weight of PEN-221 was only 2 kDa, but it was found to be rapidly cleared in circulation. To generate small-size ADCs, more efforts are needed [89].

Site-specific ADCs are a recent trend in constructing ADCs. By coupling cytotoxic payloads to engineered specific amino acids, unnatural amino acids, short peptide tags, or through glycoengineering, researchers have developed next-generation site-specific conjugation techniques [90]. In fact, some site-specific ADCs have been put into clinical trials. The anti-HER2 ADC ARX788 and the anti-ROR1 ADC NBE-002 were both site-specific ADCs, as discussed in the previous two sections. A promising technique named AJICAP brought about widespread attention. ADCs constructed with AJICAP technology exhibited increased drug tolerance and expanded therapeutic index in preclinical experiments [91]. The updated second-generation AJICAP technique further improved the stability of Fc affinity reagents, which can be used to produce more than 20 ADCs including various antibodies and payloads [92]. The site-specific ADCs were expected to be ideal next-generation ADCs with homogeneity, safety, and efficacy that are worth expecting.

Breakthroughs also may take place in using immunomodulators as payloads. One of the goals of the next-generation ADCs was to combine the immune-stimulating property of immunotherapy and the precise targeting property of antibodies, namely, immune-stimulating antibody conjugates (ISACs) [93]. ISACs significantly differ from traditional ADCs with cytotoxic payloads because of their ability to boost the immune system to kill cancer cells instead of killing cancer cells directly. Toll-like receptors (TLRs) are believed to be able to switch the TME and promote antitumor immune responses. TLR agonists can be payloads for ISACs due to their preferable tolerance. In preclinical models, an anti-HER2 ISAC with TLR7/8 antagonist effectively resulted in tumor elimination in mouse models. Surprisingly, the ISAC-mediated immunological memory can protect mice from both HER2-positive and HER2-negative tumors in the rechallenge [94]. Another anti-HER2 ISAC with TLR7 agonist confirmed the activation of antitumor immunity [95]. The application of ISACs in the clinical practice of breast cancer is expected.

## 6. Conclusions

The arise of ADCs has made great breakthroughs in cancer treatment. The ADCs approved by the FDA and the promising efficacy of ADCs in clinical trials have attracted increasing attention after three generations of development. For breast cancer, ADCs including T-DM1, T-DXd, and SG brought hope to those who suffer from advanced, metastatic tumors. Emerging ADC agents and targets are under exploration to further eradicate cancer cells precisely. To overcome challenges, the mechanism underlying adverse effects, pharmacokinetics, and drug resistance needs comprehensive comprehension, and the development and progress of ADCs are still warranted. Identifying and validating new antigens and antibodies, designing payloads with optimal toxicity, and designing linkers to balance stability and payload release appear to be significant in the field of optimizing ADCs. Explorations in site-specific ADCs and ISACs may also make breakthroughs in developing ADCs. Altogether, future ADCs for targeted cancer therapy are anticipated expectations given the continuous efforts in this field.

## Figures and Tables

**Figure 1 ijms-24-11903-f001:**
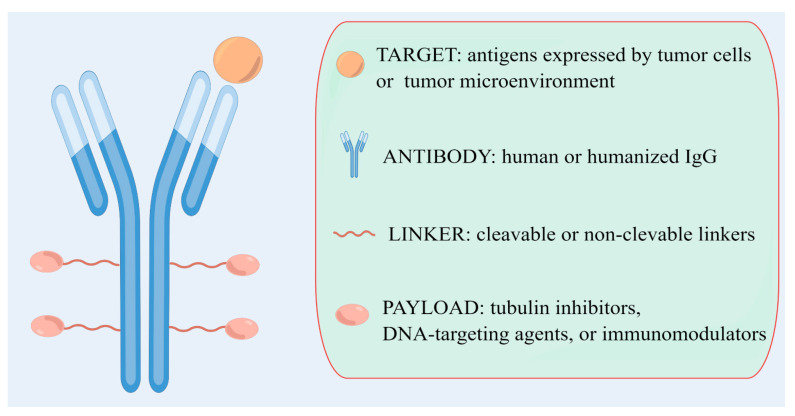
The structure of ADCs. ADC, antibody-drug conjugate.

**Figure 2 ijms-24-11903-f002:**
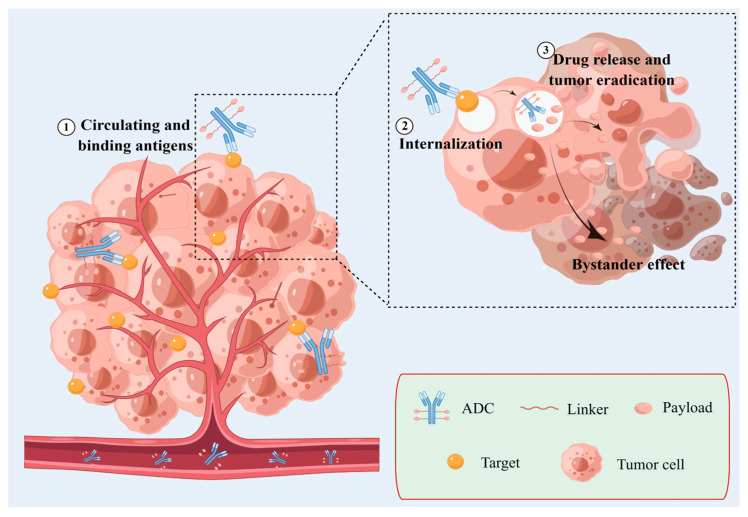
ADCs play antitumor roles by circulating and binding antigens, internalization, and drug release. ADC, antibody-drug conjugate.

**Table 1 ijms-24-11903-t001:** ADCs for breast cancer targeting HER2 and representative trials.

ADCs	Antibodies	Payloads	Representative Trials	Phase	Approval
T-DM1	Trastuzumab	DM1	EMILIA (NCT00829166)	III	Yes
			KATHERINE (NCT01772472)	III	
			ATEMPT (NCT01853748)	II	
			KATE2 (NCT02924883)	II	
			KATE3 (NCT04740918)	III	
			Astefania (NCT04873362)	III	
T-DXd	Trastuzumab	DXd	DESTINY breast-03 (NCT03529110)	III	Yes
			DESTINY breast-09 (NCT04784715)	III	
			NCT02564900	I	
			DESTINY breast-04 (NCT03734029)	III	
			DESTINY breast-07 (NCT04538742)	I/II	
			DESTINY breast-08 (NCT04556773)	Ib	
RC48	Hertuzumab	MMAE	NCT02881190	I	Yes
			NCT03052634	Ib	
SYD985	Trastuzumab	Duocarmycin	TULIP (NCT03262935)	III	Not yet
A166	Trastuzumab	Duostain-5	NCT03602079	I	Not yet
ARX788	anti-HER2 monoclonal antibody with the modified heavy chain Ala114	Amberstatin	NCT03255070	I	Not yet

**Table 2 ijms-24-11903-t002:** Emerging ADC Targets for breast cancer and representative trials.

Targets	ADCs	Antibodies	Payloads	Representative Trials	Phase	Approval
Trop2	SG	Sactizumab	SN-38	ASCENT(NCT02574455)	III	Yes
				TROPiCS-02 (NCT03901339)	III	
Trop2	Dato-DXd	Datopotamab	DXd	TROPION-PanTumor01 (NCT03401385)	I	Not yet
				TROPION Breast01 (NCT05104866)	III	
				TROPION Breast02 (NCT05374512)	III	
Trop2	SKB264	Anti-Trop2	Topoisomerase I inhibitor	NCT05347134	III	Not yet
				NCT05445908	II	
				NCT04152499	I/II	
PTK7	Cofetuzumab pelidotin	Cofetuzumab	Au0101	NCT02222922	I	Not yet
ROR1	NBE-002	Anti-ROR1	PNU-159682	NCT04441099	I/II	Not yet
ROR2	CAB-ROR2-ADC	Anti-ROR2	Undisclosed	NCT03504488	I	Not yet
B7-H3	MCG018	Anti-B7-H3	Duocarmycin	NCT03729596	I	Not yet
gpNMB	CDX-011	Glembatumumab	MMAE	EMERGE (NCT01156753)	II	Not yet
				NCT00704158	I/II	
				METRIC (NCT01997333)	II	
FRα	MORAB-202	Anti-FRα	Eribulin	NCT04300556	I/II	Not yet
HER3	HER3-DXd	Patritumab	DXd	NCT02980341	I/II	Not yet
				NCT04610528	I	
CA6	SAR566658	DS6	DM4	NCT02984683	II	Not yet

**Table 3 ijms-24-11903-t003:** Adverse effects and frequencies related to T-DM1, T-DXd, and SG.

Adverse Events	Frequency in T-DM1 [79]	Frequency in T-DXd [38]	Frequency in SG [52,80]
Thrombocytopenia	14.3%	4.3%	14%
Neutropenia	2.4%	20%	26%
Anemia	3.8%	8.6%	11%
Fatigue	2.4%	6%	9%
ILD	-	15.5% (Grade 3:1%)	-
Drop of LVEF (left ventricular ejection fraction)	2% (Grade 3: 0.7%)	1.6% (Grade 0.5%)	-
Increase in ALT	15%	-	-
Nausea	-	7.6%	6%
Vomit	-	4.3%	6%

## Data Availability

Not applicable.

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
