# Peer review of "Antibody-Drug Conjugates for Breast Cancer Treatment: Emerging Agents, Targets and Future Directions"

_ijms, 2023, doi:10.3390/ijms241511903_

Round 1
Reviewer 1 Report
This review will be very helpful.
Author Response
Dear reviewer,
We feel great thanks for your professional comments concerning our manuscript “Antibody-drug Conjugates for Breast Cancer Treatment: Emerging Agents, Targets and Future Directions”. Those comments are all valuable and very helpful for furthermore improving our paper. We read the comments carefully and have made corresponding corrections which we hope to meet with your approval.
Yours sincerely,
Tinglin Yang, Wenhui Li, Tao Huang, and Jun Zhou
All modifications in the manuscript have been marked up by using the “track changes” function in MS Word.
Comment: This review will be very helpful.
Response: Thank you for your recognition and encouragement.
Reviewer 2 Report
Thank you for the opportunity to review this manuscript on antibody-drug conjugates (ADCs) for breast cancer. While the topic is indeed timely and pertinent, I regret to say that the present manuscript does not meet the standards required for the current form of publication in this high impact journal.
The manuscript lacks originality and appears to be mostly a reiteration of clinical trial results. The points highlighted in the "Future direction" section, such as dual ADC, dual epitope, and small-sized antibody, are not novel and have been discussed in other reviews. The discrepancy between the level of the current review and what would be expected for a publication in a high impact journal is quite pronounced.
The main problem is the lack of originality. For example, there are several reviews published in 2023 that discussed ADCs for breast cancer (the references below). It is important that the authors clarify what distinguishes their review from these existing publications.
1. Antibody–drug conjugates: the evolving field of targeted chemotherapy for breast cancer treatment https://doi.org/10.1177/175883592311836
2. Antibody–Drug Conjugates in Breast Cancer: Ascent to Destiny and Beyond—A 2023 Review https://doi.org/10.3390/curroncol30070474
3. Advances in Targeted Therapy of Breast Cancer with Antibody-Drug Conjugate https://doi.org/10.3390/pharmaceutics15041242
T-DM1 and T-Dxd are indeed blockbuster ADCs. However, they also have significant side effects. It is necessary to report and analyze these side effects and suggest possible solutions or improvements. This kind of in-depth analysis is lacking in the current manuscript.
The recent emergence of immunostimulatory ADCs such as ISACs and their ongoing preclinical evaluation in breast cancer is an exciting development in the field. The authors should discuss these ADCs, clearly differentiate them from the more traditional high potency payload ADCs, and provide a relevant discussion.
Recent trends such as site-specific ADCs should also be addressed. Methods have been reported to conveniently generate HER-2-type site-specific ADCs, which may expand the therapeutic index (Mol. Pharmaceutics 2021, 18, 11, 4058-4066; Bioconjugate Chem. 2023, 34, 4, 728-738). This aspect is conspicuously absent from the review.
Furthermore, Figure 1 should preferably show an ADC with an antibody. The current figure is not sufficiently representative and does not adequately convey the complexity and structure of an ADC.
In conclusion, the authors need to significantly revise and expand their manuscript to provide a more comprehensive, unique, and in-depth review of the current state of ADCs in breast cancer therapy. Such an undertaking would greatly enhance the value of the manuscript to readers and potentially make it suitable for publication in this high-impact journal.
Author Response
Dear reviewer,
We feel great thanks for your professional comments concerning our manuscript “Antibody-drug Conjugates for Breast Cancer Treatment: Emerging Agents, Targets and Future Directions”. Those comments are all valuable and very helpful for furthermore improving our paper. We read the comments carefully and have made corresponding corrections which we hope to meet with your approval.
Yours sincerely,
Tinglin Yang, Wenhui Li, Tao Huang, and Jun Zhou
The main corrections in the paper and the response to your comments are as follows. All modifications in the manuscript have been marked up by using the “track changes” function in MS Word.
Comment #1: Thank you for the opportunity to review this manuscript on antibody-drug conjugates (ADCs) for breast cancer. While the topic is indeed timely and pertinent, I regret to say that the present manuscript does not meet the standards required for the current form of publication in this high impact journal.
The manuscript lacks originality and appears to be mostly a reiteration of clinical trial results. The points highlighted in the "Future direction" section, such as dual ADC, dual epitope, and small-sized antibody, are not novel and have been discussed in other reviews. The discrepancy between the level of the current review and what would be expected for a publication in a high impact journal is quite pronounced.
The main problem is the lack of originality. For example, there are several reviews published in 2023 that discussed ADCs for breast cancer (the references below). It is important that the authors clarify what distinguishes their review from these existing publications.
- Antibody–drug conjugates: the evolving field of targeted chemotherapy for breast cancer treatment https://doi.org/10.1177/175883592311836
- Antibody–Drug Conjugates in Breast Cancer: Ascent to Destiny and Beyond—A 2023 Review https://doi.org/10.3390/curroncol30070474
- Advances in Targeted Therapy of Breast Cancer with Antibody-Drug Conjugate https://doi.org/10.3390/pharmaceutics15041242
Response #1: Thank you for your professional work concerning our paper. We believe our manuscript is appropriate for publication in the special issue "Novel Targeted Therapies in Cancer" of the International Journal of Molecular Sciences because our review provides clinicians and patients with a thorough summarization of existing and emerging targets and ADCs in targeted breast cancer therapy. With the success T-DM1 and T-DXd have achieved, combinations of ADCs and immunotherapy were under exploration. We discussed related clinical trials in our revised manuscript (Line 181-216, Page 4-5), (Line 245-255, Page 5), which was hardly mentioned in previous reviews.
In the section “5.2 Future Directions”, we mainly developed ADCs constructed with new trends of designs that have not been widely practiced in clinical. Dual targets, dual payloads, and reduced-size ADCs are indeed fashion trends in developing ADCs, whose clinical potential has not been tested. Besides, we supplemented other two future directions including site-specific ADCs and immune-stimulating antibody conjugates (ISACs) according to your comments (Line 558-592, Page 11-12).
The originality of our paper is discussed as follows.
- Antibody–drug conjugates: the evolving field of targeted chemotherapy for breast cancer treatment. Compared with this review, our manuscript concluded the structure and evolution generations of the ADCs. We also discussed the challenges and future directions of ADCs, which were not mentioned in this review. Furthermore, we gave a more competent insight into ADC targets beyond HER2 (eight more targets) to clinicians in breast cancers. Novel trials of ADCs plus immunotherapy were also concluded in our manuscript.
- Antibody–Drug Conjugates in Breast Cancer: Ascent to Destiny and Beyond—A 2023 Review. This review summarized the current status of T-DM1, T-DXd, and SG in breast cancer. However, significant insufficiency lies in the discussion of ADCs in clinical trials by the year 2023 because only Dato-DXd and trastuzumab duocarmazine was talked about, whereas it was thoroughly discussed in our review. Compared with the review that only discussed HER2 and Trop2, we reviewed 7 potential targets beyond them. Besides, the evolution of ADC structure that concluded in our manuscript contributes to further understanding the future directions of ADCs. Combinations with immunotherapy were also illustrated.
- Advances in Targeted Therapy of Breast Cancer with Antibody-Drug Conjugate. Only LIV-1 as a target beyond HER2 and Trop2 was mentioned in the table of this review. Structure evolutions of ADC, more promising targets, and updated future directions made our manuscript different from this review. We also reviewed the regimen of ADCs plus immunotherapy.
Therefore, our manuscript not only reviewed approved ADCs in breast cancer, but also gave the latest insights into adequate potential targets, structures and evolutions of ADCs, and updated future directions. Besides, innovations in this review were also demonstrated by the discussion of novel combined regimens of ADCs and immunotherapy. As our title described, our originality lies in emerging agents, targets, and future directions of ADCs for breast cancer treatment.
Comment #2: T-DM1 and T-Dxd are indeed blockbuster ADCs. However, they also have significant side effects. It is necessary to report and analyze these side effects and suggest possible solutions or improvements. This kind of in-depth analysis is lacking in the current manuscript.
Response #2: T-DM1 and T-Dxd are the most widely used ADCs in breast cancer, and it is necessary to comprehend possible adverse effects. We summarized the common adverse effects of ADCs in the section “5.1 Challenges in ADCs” in our previous manuscript. In this revised version, we further clearly interpreted the specific adverse effects of T-DM1 and T-DXd in this section, along with mechanisms, management, and frequencies of occurrence. Further, as the other approved ADC in breast cancer, adverse events of SG were also supplemented (Line 491-511, Page 10). We also added Table 3 (Line 511, Page 10-11) to show the adverse effects and frequencies of T-DM1, T-DXd, and SG concretely.
Comment #3: The recent emergence of immunostimulatory ADCs such as ISACs and their ongoing preclinical evaluation in breast cancer is an exciting development in the field. The authors should discuss these ADCs, clearly differentiate them from the more traditional high potency payload ADCs, and provide a relevant discussion.
Response #3: Novel ISACs provide a new trend in ADC's future directions. It showed strong immune-stimulating functions in mouse models. Besides, it facilitates the host with immunological memory beyond the target. We added related discussions in section 5.2 “Future directions”, distinguishing it from traditional ADCs and giving more information about next-generation ADCs (Line 580-592, Page 12).
Comment #4: Recent trends such as site-specific ADCs should also be addressed. Methods have been reported to conveniently generate HER-2-type site-specific ADCs, which may expand the therapeutic index (Mol. Pharmaceutics 2021, 18, 11, 4058-4066; Bioconjugate Chem. 2023, 34, 4, 728-738). This aspect is conspicuously absent from the review.
Response #4: Thank you for your replenishment. Site-specific is indeed another future direction of ADCs. In fact, several site-specific ADCs have been put into clinical investigations, including the ARX788 discussed in section 3.3 (Line 253-262, Page 6) and the NBE-002 discussed in section 4.3 (Line 333-338, Page 8) of our manuscript. We supplemented several coupling approaches of site-specific ADCs in section 5.2 “Future directions”. The AJICAP technique mentioned in your comment demonstrated great progress in improving the therapeutic index, and we have discussed the technique and quoted papers mentioned in the comment (Line 558-579, Page 11-12).
Comment #5: Furthermore, Figure 1 should preferably show an ADC with an antibody. The current figure is not sufficiently representative and does not adequately convey the complexity and structure of an ADC.
Response #5: Thank you for your careful review. We replotted Figure 1 (Line 65-66, Page 2) to exhibit a detailed structure of ADCs, including antibodies, linkers, and payloads. Besides, we augmented Figure 2 (Line 104-105, Page 3) to further demonstrate the specific antitumor mechanism ADCs play.
Comment: In conclusion, the authors need to significantly revise and expand their manuscript to provide a more comprehensive, unique, and in-depth review of the current state of ADCs in breast cancer therapy. Such an undertaking would greatly enhance the value of the manuscript to readers and potentially make it suitable for publication in this high-impact journal.
Response: We have made major revisions including adding the combination of ADCs and immunotherapy, adverse effects of T-DM1, T-DXd, and SG, ISACs, and site-specific ADCs. Minor grammatical errors and typos were also carefully revised. Thank you for your comments again and we hope the revised manuscript could meet with your approval.

Reviewer 3 Report
This article provides a comprehensive review of antibody-drug conjugates (ADCs) used in the treatment of breast cancer. The review encompasses both established and emerging agents, highlighting different targets that these ADCs aim to address. The article also discusses the challenges encountered in the field of ADC development and explores potential future directions for improvement. Clinical trials showcasing representative examples of ADCs in breast cancer treatment are examined, along with an exploration of the structure and evolution of ADCs. This is an interesting review and is very important review in therapeutic aspect of breast cancer.
Minor Comments:
Authors should include introduction how challenging it is to target ER+ and triple negative breast cancer tumors and also mention the caveats to target these tumors using ADCs.
Authors should include how ADC's are advantages when compared to combination of antibodies. For example as shown in these papers https://pubmed.ncbi.nlm.nih.gov/35895872/
https://pubmed.ncbi.nlm.nih.gov/29212784/
Author Response
Dear reviewer,
We feel great thanks for your professional comments concerning our manuscript “Antibody-drug Conjugates for Breast Cancer Treatment: Emerging Agents, Targets and Future Directions”. Those comments are all valuable and very helpful for furthermore improving our paper. We read the comments carefully and have made corresponding corrections which we hope to meet with your approval.
Yours sincerely,
Tinglin Yang, Wenhui Li, Tao Huang, and Jun Zhou
The main corrections in the paper and the response to your comments are as follows. All modifications in the manuscript have been marked up by using the “track changes” function in MS Word.
Comment #1: Authors should include introduction how challenging it is to target ER+ and triple negative breast cancer tumors and also mention the caveats to target these tumors using ADCs.
Response #1: Thank you for your comment. TNBC lacks molecular targets to treat, and endocrine therapy is the first-line therapy for ER+ breast cancers rather than targeted therapy. Despite it was challenging to target TNBCs, certain progress has been made in this field. An ADC targeting Trop2 named sacituzumab govitecan (SG) has been approved for metastatic TNBCs. Detailed explanations were added in section 4 (Line 304-311, Page 7) as follows:
With the great progress of ADCs targeting HER2, efforts were taken to extend the application of ADCs in other subtypes of breast cancers. However, TNBC has long been known as a hard-to-cure subtype due to the lack of molecular markers [49]. For hormone receptor (HR)-positive breast cancers, endocrine therapy, rather than targeted therapy was recommended as the first-line therapy [50]. It was in doubt whether there existed potential targets for ADC treatment in HR-positive breast cancers and TNBCs. To a-dress the efficacy of ADCs in these subtypes, as well as explore the possibility of improving therapeutic effects in HER2-positive breast cancers, more targets are being in-vestigated in breast cancer.
Comment #2: Authors should include how ADC's are advantages when compared to combination of antibodies. For example as shown in these papers https://pubmed.ncbi.nlm.nih.gov/35895872/ https://pubmed.ncbi.nlm.nih.gov/29212784/
Response #2: ADCs showed the potential to overcome resistance to antibodies due to the combination of cytotoxic payloads, and related discussions were added in section 1 (Line 35-39, Page 1) as follows:
For drug resistance to targeted therapy with monoclonal antibodies, the activation of alternative pathways is a common mechanism8. Compared with the strategies of combining antibodies, ADCs that facilitate both antibody-mediated effects and cytotoxic payloads provide preferable efficacy by the addition of different antitumor effects.

Round 2
Reviewer 2 Report
I appreciate the authors' careful amendments. I believe the current form deserves to be published.